# *Megalurothrips usitatus* Directly Causes the Black-Heads and Black-Tail Symptoms of Cowpea along with the Production of Insect-Resistance Flavonoids

**DOI:** 10.3390/plants12223865

**Published:** 2023-11-15

**Authors:** Yunchuan He, Yang Gao, Hainuo Hong, Jiamei Geng, Qiulin Chen, Ying Zhou, Zengrong Zhu

**Affiliations:** 1Hainan Institute, Zhejiang University, Yazhou District, Sanya 572025, China; heyunchuan@zju.edu.cn (Y.H.); gyon@zju.edu.cn (Y.G.); 12216111@zju.edu.cn (H.H.); 22116243@zju.edu.cn (J.G.); 22116247@zju.edu.cn (Q.C.); zrzhu@zju.edu.cn (Z.Z.); 2State Key Laboratory of Rice Biology, Institute of Insect Sciences, Zhejiang University, Hangzhou 310058, China

**Keywords:** *Megalurothrips usitatus*, thrip, cowpea, pathogen, flavonoid, nutrition

## Abstract

The thrip (*Megalurothrips usitatus*) damages the flowers and pods of the cowpea, causing “black-heads and black-tails” (BHBT) symptoms and negatively affecting its economic value. However, the mechanism by which BHBT symptoms develop is still unknown. Our results showed that the microstructure of the pod epidermis was altered and the content of the plant’s resistance-related compounds increased after a thrip infestation. However, the contents of protein and free amino acids did not change significantly, suggesting that the nutritional value was not altered. Pathogens were found not to be involved in the formation of BHBT symptoms, as fungi and pathogenic bacteria were not enriched in damaged pods. Two herbivory-induced flavonoids—7,4′-dihydroxyflavone and coumestrol—were found to exert insecticidal activity. Our study clarified that BHBT symptoms are directly caused by the thrip. Thresholds for pest control need to be reconsidered as thrip herbivory did not degrade cowpea nutrition.

## 1. Introduction

Cowpea (*Vigna unguiculata* L. (Walp.)), a dicotyledonous crop belonging to the family Fabacea, is among the most important pulse crops grown worldwide. Cowpea is cultivated across the majority of tropical and sub-tropical countries as a vegetable, fodder, and seed crop [1]. Cowpeas are a vital source of plant-based protein (21–40%) and essential nutrients, including potassium (K, 19,743.74 mg/kg), calcium (Ca, 4175.56 mg/kg), and iron (Fe, 71.43 mg/kg) [2]. Cowpeas also contain phenolic compounds, such as flavonoids, phenolic acids, proanthocyanidins, and anthocyanins [3], which promote health by acting as antioxidants.

However, several insect pests attack the crop from the seedling stage until harvest, including *Aphis craccivora* Koch (Hemiptera: Aphididae) [4], *Callosobruchus maculatus* (Coleoptera: Chrysomelidae) [5], *Nemorilla maculosa* Meigen (Diptera: Tachinidae) [6], and *Megalurothrips usitatus* Bagnall (Thysanoptera: Thripidae) [7]. Among these, the bean flower thrip (*M. usitatus*) is one of the most serious pests of cowpea in many Chinese provinces, especially Hainan [8]. During feeding and oviposition, *M. usitatus* can pierce leaves, flowers, and pods, causing leaf wrinkling, growth point atrophy, necrosis, premature bud- and flower-drop, and pod scab [9]. In addition, *M. usitatus* can also transmit the tobacco streak virus, which can result in severe crop damage [10].

Currently, some studies have been performed into the physiological and biochemical changes associated with *Megalurothrips sjostedti* in cowpea. For example, amino acids, reducing sugars, total protein, glucose content, total phenol, tannin, terpenoids compounds, aglycones of flavones and flavonols and antioxidant activity cowpea floral buds in relation to resistance to *M. sjostedti* [11]. No studies have been reported on the harm of *M. usitatus* on cowpea. In cowpea, the pods have both the highest economic and nutritional value. Pod set and filling occur through the transportation of photoassimilates from leaves to fruits [12]. Attack by *M. usitatus* can result in reduced yield and economic value, as thrip damage to immature pods can result in surface roughening, deformation, brown discoloration, rust spots, and pod scab, as well as “black-heads and black-tails” (BHBT) symptoms [9]. However, the mechanism of BHBT is poorly understood, and it is unknown whether BHBT is caused directly by thrips or indirectly by pathogenic fungi or bacteria. It is also unknown whether thrip damage induces the biosynthesis of anti-herbivory compounds or induces changes in epidermal tissue structure or physiology in cowpea pods. Finally, although BHBT is well known to be responsible for economic losses, it is unknown whether BHBT changes the nutritional value of the affected cowpeas. An increasing number of studies have shown that inducing the resistance of plants against herbivory is a promising method for pest control. The metabolomics is a classical technique to explore plant defense mechanisms, and numerous studies have been conducted [13].

In order to fill these knowledge gaps, the symptoms of “BHBT” caused by thrips were studied by analyzing the epidermal morphology, plant-resistance-related compounds, microbial community composition, and secondary metabolomics of cowpea pods. The results suggest that thrip herbivory directly causes BHBT symptoms, without the involvement of pathogens. Furthermore, herbivory by thrips resulted in the upregulated biosynthesis of anti-herbivory compounds, such as flavonoids and isoflavonoids. Finally, although herbivory by thrips altered the morphology of the pods, their nutritional value was minimally affected.

## 2. Results

### 2.1. M. usitatus Alters Cowpea Pod Epidermal Morphology

Laboratory tests showed that BHBT symptoms can result from *M. usitatus* eating healthy pods (Appendix A). The microstructural differences between healthy pods (CK) (Figure 1A) and BHBT pods (T) (Figure 1B–D) were analyzed. Analyses of fresh sections, paraffin-embedded sections, and SEM images revealed significant morphological and structural difference between CK and T pods. The pod epidermis (EP) of the CK was yellow-green, with an organized texture (Figure 1E) and normally-shaped epidermal cells which appeared blue after staining (Figure 1I). Conversely, the EP of the T was lignified and brown (Figure 1F,H), with severely lignified epidermal cells which appeared red after staining (Figure 1J). The pod parenchyma (PA) hierarchical structure is clear in T (Figure 1H). CK treatment epidermis was smooth and yellow–green (Figure 1G), and the pod epidermis was composed of a layer of cells with a tight arrangement and regular cell morphology (Figure 1K). However, the epidermal cells were broken and disordered in T (Figure 1L). Both CK and T PA and vascular bundle (VB) cells were tightly packed, with similar cell morphology and no obvious damage (Figure 1K,L). The SEM analysis showed that CK contained more stomata, and a larger stomatal gap opening, than T (Figure 1M,N). Analysis of longitudinal sections showed that the epidermal cells were square and round in CK (Figure 1O), but narrow and oval in T (Figure 1P). In CK, the pods had regular ridge lines with clear tissue texture (Figure 1M,O). However, the ridge lines were disordered and scabbed in T due to thrip damage, and lacked normal tissue texture (Figure 1M,O). Notably, analyses of the fresh sections, paraffin-embedded sections, and SEM images did not result in the identification of fungal hyphae in the pod epidermis, suggesting that fungi may not be involved in the formation of BHBT symptoms.

### 2.2. M. usitatus Alters Cell Wall Components and Stress Indicators

Because *M. usitatus* was found to damage the pod epidermis and alter the cell wall structure, we further analyzed herbivory-associated changes in cell wall components and stress indicators. Herbivory by *M. usitatus* significantly increased the contents of cellulose (from 850.07 mg/g to 934.72 mg/g, *t*_8_ = 2.5414, *p* = 0.0346), lignin (from 228.50 mg/g to 267.73 mg/g, *t*_8_ = 2.9246, *p* = 0.0344), and pectin (from 3.58 mg/g to 5.70 mg/g, *t*_8_ = 4.4108, *p* = 0.0023) (Figure 2A–C). This was consistent with the observed changes in cell wall morphology. In addition, the contents of callose (from 57.59 ng/g to 262.67 ng/g, *t*_8_ = 8.7819, *p* = 0.0001) and ROS (from 855.00 pg/g to 2001.06 pg/g, *t*_8_ = 9.9118, *p* = 0.0001) increased significantly following *M. usitatus* (Figure 2D,E), suggesting a strong stress response in response to herbivory. Furthermore, herbivory by *M. usitatus* significantly increased the contents of ABA, JA, and JA-Ile. Specifically, there were significant differences in ABA (t8 = 50.6861, *p* = 0.0001) (Figure 2F), JA (*t*_8_ = 20.1275, *p* = 0.0001) (Figure 2G), and JA-Ile (*t*_8_ = 5.9956, *p* = 0.0039) (Figure 2H) between T and CK, but no significant differences in SA (*t*_8_ = 1.9727, *p* = 0.1190) (Figure 2I).

### 2.3. M. usitatus does Not Reduce the Protein or Free Amino Acid Content of Cowpea Pods

The primary nutritional value of cowpea is to provide high-quality protein. To determine whether herbivory by *M. usitatus* affects the nutritional value of cowpea, the contents of protein and free amino acids were analyzed. The results showed that the protein content was not significantly different between T and CK (*t*_8_ = 0.2410, *p* = 0.8156) (Figure 3). Moreover, there was no significant difference in the contents of most of the free amino acid (Table 1), with only the content of Ala increased in T. Taken together, these results indicate that herbivory by *M. usitatus* induces BHBT symptoms but does not appreciably alter the nutritional value of cowpea.

### 2.4. Pathogenic Bacteria Were Not Enriched in M. usitatus-Damaged Cowpea Pods

To investigate whether pathogenic bacteria are involved in the development of BHBT symptoms, 16S rRNA amplicon sequencing was performed. According to the results of the OUT-level PCA analysis, the bacterial communities under the two treatments were clearly separated and the three biological replicates of each treatment were clustered together. These results indicate that the analysis was reproducible and reliable, and that there were significant differences between the two treatments (Figure 4A). According to the results of the OUT-level alpha diversity analysis, the Shannon diversity index was significantly different (*p* = 0.004717) between CK and T (Figure 4B), indicating that CK pods contained a more diverse bacterial community. No significant differences were observed in the Ace (*p* = 0.1333) and Chao (*p* = 0.1245) indices between CK and T (Figure 4C). Although the relative abundances of different bacterial genera were significantly different between CK and T (Figure 4D,E), no pathogenic bacteria were detected by 16S rRNA amplicon sequencing. These results suggest that while herbivory by *M. usitatus* can alter the bacterial community of cowpea epidermis, pathogenic bacteria are not involved in the development of BHBT symptoms.

### 2.5. M. usitatus Induced the Biosynthesis of Anti-Herbivory Metabolites

Plants experiencing insect herbivory often respond by upregulating the biosynthesis of anti-herbivory metabolites. Here, we studied changes in anti-herbivory metabolites in cowpea pods after *M. usitatus* attack. A total of 779 metabolites were identified in CK and T pods, including 386 flavonoids (49.55%), 195 phenolic acids (25.03%), 107 alkaloids (13.74%), 33 terpenes (4.24%), 28 lignans and coumarins (3.59%), 8 quinones (1.03%), and 22 other metabolites (2.82%) (Figure 5A). A total of 583 DEMs were identified between CK and T, accounting for 74.8% of the total metabolites (Appendix A). CK pods contained 48 unique metabolites (6.2%) and T pods contained 148 unique metabolites (19%) (Figure 5B). The two treatments were clearly divided into two groups (Figure 5C), and the contents of most compounds increased after *M. usitatus* (Figure 5D). The isoflavonoid and flavonoid biosynthesis pathway was found to be most responsive to *M. usitatus* (Figure 5E,F). Isoliquiritigenin, liquiritigenin, butin, and 7,4′-dihydroxyflavone, which take part in the flavonoid biosynthesis pathway, were significantly upregulated. In the isoflavonoid biosynthesis pathway, daidzein, 2′-hydroxydaidzein, isoformononetin, formononetin, calycosi, and coumestrol were highly upregulated.

Many anti-herbivory and anti-insect compounds are derived from flavonoids and isoflavones. To determine whether *M. usitatus* induces the formation of insecticidal metabolites in cowpea pods, the two most significantly upregulated compounds (7,4′-dihydroxyflavone and coumestrol) were tested for insecticidal activity against *M. usitatus*. Both compounds significantly increased the mortality of *M. usitatus* adults (*t*_18_ = 3.4757, *p* = 0.0027 for 7,4′-dihydroxyflavone and *t*_18_ = 2.7440, *p* = 0.0179 for coumestrol) (Figure 5G).

## 3. Discussion

### 3.1. BHBT Symptoms Are Caused Directly by M. usitatus

Because plants are the primary producers in food webs, they attract a variety of heterotrophs, including herbivorous insects and phytopathogenic microbes. Both of these groups rely on plants as their sole source of nutrition and can cause devastating effects. To counteract these effects, plants have evolved an intricate defense system of chemical and physical barriers, including herbivory-induced changes to specialized morphological structures such as plant cell walls. Analyses of paraffin-embedded sections and SEM images revealed significantly different morphological structures in cross sections and vertical sections of T and CK pods. Pods with BHBT symptoms are characterized by a brown epidermis, negatively affecting photosynthesis and seed development [14]. This is because the pod cell wall plays a crucial role in regulating carbon partitioning during seed filling [12,15]. Previous research has demonstrated that the legume pod cell wall is photosynthetic and contributes to seed yield in many crops, including alfalfa (*Medicago sativa*), chickpea (*Cicer arietinum*), soybean (*Glycine max*), and lentil (*Lens culinaris*) [15].

Infestation by fungi or bacteria often results in morphological changes in the affected plant. For example, fungal diseases such as rusts and smuts can cause discoloration and deformation of leaves, stems, and flowers. Rust-infected leaves may develop yellow or reddish-orange pustules, while smut-infected flowers may become blackened or distorted [16]. Laboratory tests showed that BHBT symptoms can result from *M. usitatus* of healthy pods (Appendix A). However, it was unclear whether pathogenic fungi or bacteria participate in the formation of BHBT symptoms. We did not observe hyphae in our SEM images, suggesting that fungi are not involved in the development of BHBT symptoms. The 16S rRNA sequencing results indicated that while herbivory by *M. usitatus* can alter the bacterial community of cowpea epidermis, pathogenic bacteria are not involved in the development of BHBT symptoms. Taken together, these results suggest that herbivory by *M. usitatus* directly results in the formation of BHBT symptoms in cowpea, and this process does not involve the participation of pathogenic fungi or bacteria.

### 3.2. M. usitatus Altered Stress-Responsive Markers but Not Nutritional Value

Compared to CK, the contents of ROS, callose, cellulose, lignin, and pectin were significantly higher in T pods. These results confirm that cell wall components act as barriers to limit herbivory by insects. ROS play an important role in plant defense against biotic stressors, including herbivorous insects. Plants may even generate ROS in response to insect eggs, thus effectively fighting against future larval herbivory [17]. Callose deposition is a well-documented plant defense response to pathogen and insect attacks, and this defense response may be particularly relevant to phloem-based defense against pathogens and piercing–sucking insects [18]. Cell wall lignification may be induced when plants are wounded or become infected by pathogens, suggesting that lignin may act as a chemical or physical barrier to protect the rest of the plant cell from further damage [19]. Furthermore, lignin content is positively correlated with tissue toughness and tissue toughness mediates herbivore damage in plants [20]. Tissues with high lignin content are less palatable than tissues with low lignin content, possibly due to the decreased digestibility of the strengthened cell walls [20]. Increased lignin deposition might have additional negative effects on insect fitness because phenoloxidase enzymes are involved not only in lignin polymerization but also the generation of toxic by-products such as ROS, quinones, and peroxides [21]. Cellulose content is positively correlated with resistance to pests [22], and increased cellulose and lignin content results in increased fiber content [23]. Overall, it appears that the likely mechanism of plant cell wall-associated protection against herbivory likely results from a combination of the toxic by-products of lignin synthesis (i.e., resulting from phenoloxidase activity), ROS, callose, cellulose, and pectin.

The processes of senescence and response to insect damage are closely linked to the activity of phytohormones such as ABA, JA, and SA [24]. Here, we found that the contents of ABA, JA, and JA-Ile were significantly increased by *M. usitatus*. Both JA and JA-Ile are major signals in the insect defense response [10], and ABA has been found to play a role in the insect defense response in *Arabidopsis* [25]. The increase in ABA may also be induced by water loss resulting from *M. usitatus*, as ABA is also an indicator of water stress. Li et al. [26] have shown that defensive resistance of cowpea control *M. usitatus* is mediated by JA insect damage. However, there was no significant difference in SA content between treatments. This may have been due to the fact that, while SA plays an important role in the plant response to pathogens [27], we found that pathogenic microbes were not involved in the formation of BHBT symptoms in cowpea.

Cowpeas are primarily prized for their high protein content. Here, we found that herbivory by *M. usitatus* did not result in changes to either the protein or free amino acid contents. These results indicate that herbivory by *M. usitatus* does not appreciably alter the nutritional value of cowpea, although the market value will still be affected due to the BHBT symptoms.

### 3.3. M. usitatus Induced the Biosynthesis of Anti-Herbivory Metabolites

Metabolomics can be a complementary tool to understand the biotic stress response in plants. The metabolomic response of leguminous crops to aphid infestation has been studied in red clover, pea, and alfalfa, and the triterpene, flavonoid, and saponin pathways were found to be responsive to aphid attack [28]. Here, we found that herbivory by *M. usitatus* primarily upregulated the flavonoid and isoflavonoid biosynthesis pathways. This result was consistent with a recent study of thrip-infested alfalfa by Zhang et al. [29]. Many flavonoids and isoflavonoids are toxic to insects, bacteria, and fungi [30]. Plant isoflavonoids exhibit a range of biological activities, acting as antimicrobial phytoalexins, inducers of the nodulation genes of symbiotic rhizobium bacteria, stimulators of fungal spore germination, insect anti-feedants, and allelochemicals.

We found that two flavonoids, 7,4′-dihydroxyflavone and coumestrol, were induced dramatically by *M. usitatus*, and both exhibited insecticidal activity: 7,4′-dihydroxyflavone is found in many plants, including Primula vulgaris and Glycyrrhiza glabra, and has been studied for its anti-inflammatory, antioxidant, and neuroprotective properties. There has also been research on the potential insecticidal properties of 7,4′-dihydroxyflavone. For example, a study by Pineda et al. [31] investigated the effect of 7,4′-dihydroxyflavone on the fall armyworm (*Spodoptera frugiperda*), a devastating pest for many crops. The researchers found that 7,4′-dihydroxyflavone significantly reduced the development, survival, and pupal weight of the fall armyworm, indicating that the compound may have insecticidal properties. Another study by Gao et al. [32] investigated the effect of 7,4′-dihydroxyflavone on the growth and development of the brown planthopper (*Nilaparvata lugens*), a pest of rice. The researchers found that 7,4′-dihydroxyflavone significantly reduced the feeding activity and survival rate of the brown planthopper, suggesting that the compound may have potential as an insecticide for rice pest management. Coumestrol, a well-known potent phytoestrogen in soybean, is a trace isoflavonoid belonging to the coumestan family which is derived from the precursor daidzein. Coumestrol acts as a phytoalexin under stress conditions, and its content has been found to increase during drought, germination, fungal infection, and exposure to chemical compounds [33]. For instance, coumestrol acts as a deterrent to the scarab beetle *Heteronychus arator* (F.) [34]. Coumestrol is antixenosic to *Epilachna varivestis* at a concentration of between 0.9 and 1.8 μg/leaf disk [35]. Lozovaya et al. [36] reported the accumulation of various isoflavonoid phytoalexins, including coumestrol, in soybean hairy roots following *F. solani f.* sp. Glycines infection. Given these results, future investigations should focus on the potential use of 7,4′-dihydroxyflavone and coumestrol as eco-friendly insecticides.

## 4. Materials and Methods

### 4.1. Plant Materials

Cowpea plants, variety ‘Yousheng 208′ (Jiangxi Huanong Seed Industry Co., Ltd., Jiangxi, China), were field grown in Bailagen (longitude 109°147486′ E, latitude 18°360859′ N, altitude 6.2 m), Shuinan village, Yacheng town, Sanya city, Hainan province, China. Cowpea pods with and without BHBT symptoms were collected at the full-grain stage.

### 4.2. Morphological Analysis

The fresh pod samples were cut transversely (~1 cm^2^) and longitudinally (~2 mm).

#### 4.2.1. Light Microscopy

The processed samples were carefully placed onto glass slides with tweezers. To each slide was added 1 drop of distilled water. Each slide was covered with a glass cover slip and observed using an OLYMPUS stereoscope (Model: BX53).

#### 4.2.2. Paraffin Sectioning

The processed samples were placed in RNase-free fixative solution (Beijing Coolaber Technology Co., Ltd., Beijing, China), vacuumized until no gas was generated and the material sank completely, and subsequently stored and fixed at 4 °C for 24 h. For dewaxing and dehydrating, the sections were placed in xylene for 20 min, and then in new xylene for 20 min, in anhydrous ethanol for 5 min, and then new anhydrous ethanol for 5 min, in 75% alcohol for 5 min, and finally washed with tap water. For safranin staining, the slices were dyed in plant safranin staining solution for 2 h, and then washed with tap water to remove excess dye. For decolorization, the slices were sequentially placed in 50%, 70%, and 80% gradient alcohol for 3–8 s each. For fast green staining, the sections were dyed in plant fast green staining solution (Zhuhai Besuo Biotechnology Co., Ltd., Zhuhai, China) for 6–20 s, and dehydrated in three tanks of absolute ethanol. To apply the transparent cover, the slices were placed in clean xylene for 5 min, and then covered with neutral gum (Beijing Coolaber Technology Co., Ltd., Beijing, China). Finally, the slices were used to conduct microscopic examination, image acquisition, and analysis.

#### 4.2.3. Scanning Electron Microscopy

The samples were further processed for scanning electron microscopy (SEM) as follows. For double fixation, the specimens were first fixed with 2.5% glutaraldehyde in phosphate buffer (pH 7.0) for at least 4 h, washed three times in phosphate buffer, postfixed with 1% OsO_4_ in phosphate buffer (pH 7.0) for 1 h, and washed three times in phosphate buffer. The specimens were first dehydrated by a graded series of ethanol (50%, 70%, 80%, 90%, 95%, and 100%) for approximately 15 to 20 min per step. Finally, the specimens were dehydrated in a LEICA EM CPD 300 critical point dryer with liquid CO_2_. The dehydrated specimens were coated with platinum–palladium and observed using a Hitachi Model SU8010FE-SEM.

### 4.3. Physiological and Biochemical Analyses

Reactive oxygen species (ROS) and callose content were detected using commercial enzyme-linked immunoassay kits (Cat. MM-072402, 48T, and Cat. LB8029A, 48T, Jiangsu Enzyme Immunoassay Industry Co., Ltd., Nanjing, Jiangsu, China). Lignin content was determined using a commercial lignin content assay kit (Cat. AKSU010U, 60T/50S, Beijing Boxbio Science & Technology Co., Ltd., Beijing, China). Cellulose content was determined using a commercial cellulose content assay kit (Cat. YX-C-B634, 50T/48S, Suzhou Comin Biotechnology Co., Ltd., Suzhou, China). Total pectin content was determined using a commercial total pectin content kit (Cat. YX-C-A517, 50T/24S, Suzhou Comin Biotechnology Co., Ltd., Suzhou, China). Protein content was determined using an enhanced BCA protein assay kit (Cat. P0010, 500T, Beyotime Biotechnology Co., Ltd., Shanghai, China). All assays were conducted according to the manufacturer’s instructions using five biological replicates.

### 4.4. Free Amino Acid Analysis

Standard configuration: A single standard mother liquor was prepared using a specific amount of standard product. An appropriate amount of each mother liquor was used to make a mixed standard product, each of which was diluted to an appropriate concentration with methanol to make a working standard solution. Both the mother liquor and working standard solution were stored at −80 °C.

Sample extraction: Samples were ground to a powder with a grinder. To each 100 mg sample was added 1000 μL of extraction buffer (methanol: acetonitrile: water = 2: 2: 1, *v*/*v*), and the mixture was shaken to dissolve the powder. The samples were ultrasonicated in an ice water bath for 10 min and frozen in liquid nitrogen for 1min, with the procedure repeated three times. Subsequently, the samples were frozen at −20 °C for 1 h. After centrifugation at 13,000 rpm for 15 min at 4 °C, the supernatant was collected and dried with a nitrogen blower. Next, 100 μL acetonitrile: water (1: 1, *v*/*v*) was added to redissolve the dried supernatant. The mixture was shaken for 30 s and ultrasonicated in an ice water bath for 10 min. After centrifugation at 13,000 rpm for 15 min at 4 °C, the supernatant was collected and prepared for further analysis.

Chromatographic conditions: We used an ACQUITY UPLC^®^ BEH C_18_ column (2.1 × 100 mm, 1.7 μm, Waters, MA, USA), with a sample size of 5 μL, a column temperature of 40 °C, a flow rate of 0.3 mL/min, and a mobile phase consisting of 100% H_2_O containing 25 mm CH_3_COONH_4_ and 25 mm NH_4_OH (A) and 100% ACN (B). The gradient elution conditions were: 0–1 min, 85% B; 1–12 min, 65% B; 12–12.1 min, 40% B; 12.1–15 min, 40% B; 15–15.1 min, 85% B; 15.1–20 min, 85% B. An electrospray ionization (ESI) source was used, with an ion source temperature of 600 °C, an ion source voltage of −4500 V, air curtain gas at 20 psi, and atomization gas and auxiliary gas at 60 psi. Scanning was performed using multiple response monitoring (5600 Q-TOF, SCIEX, Framingham, MA, USA). All experiments were conducted using five biological replicates.

### 4.5. Plant Hormone Analysis

Five replicated cowpea pods were used for analysis of abscisic acid (ABA), jasmonic acid (JA), jasmonoyl–isoleucine (JA-Ile), and salicylic acid (SA). A 0.3 g sample was extracted with 3 mL ethyl acetate containing 20 ng of D_6_-ABA, D_5_-JA, and D_4_-SA as internal standard. Then it was vortexed for 30 s and sonicated for 20 min on ice. After centrifuging at 4 °C, 12,000× *g* for 5 min, 2.5 mL supernatant was transferred into a new tube and dried under N_2_. The residue was redissolved with 200 μL methanol. The samples were analyzed by HPLC-QTOF-MS/MS (AB SCIEX, X500R). Then, 2 μL of each sample was injected into a C_18_ column (2.6 µm, 100 × 2.1 mm) (00D-4462-AN Kinetex, Phenomenex, Torrance, CA, USA). A mobile phase composed of solvent A (0.25‰ aqueous phase/formic acid) and solvent B (0.25‰ acetonitrile/formic acid) was used in a gradient mode for the separation. Negative electrospray ionization mode was used for detection.

### 4.6. Secondary Metabolome Analysis

Approximately 200 mg of freeze-dried pods was homogenized in liquid nitrogen and subjected to secondary metabolome analysis (Metware Biotechnology Co., Ltd., Wuhan, China). The sample extracts were analyzed using a UPLC-ESI-MS/MS system (UPLC, Nexera X2, Shimadzu, Kyoto, Japan; ESI-MS/MS, 4500 Q TRAP, SCIEX, MA, USA). Metabolites were identified using information from public metabolite databases and the Metware database (Metware Biotechnology Co., Ltd., Wuhan, China). All identified metabolites were subjected to principal component analysis (PCA), and significant differences were determined by setting the variable importance in projection (VIP) to ≥1 and log2 |fold-change| ≥ 1. Hierarchical clustering was performed using heatmaps. KEGG enrichment analyses of the differentially expressed metabolites (DEMs) were performed using clusterprofiler (Version 4.8.3). All experiments were conducted using three biological replicates.

### 4.7. Illumina MiSeq Sequencing and 16S rRNA Data Processing

Microbial community genomic DNA was extracted from cowpea pods using the E.Z.N.A. soil DNA kit (Omega Bio-tek, Norcross, GA, USA), according to the manufacturer’s instructions. The genomes extracted from each sample were pooled into a single test tube, checked using 1% agarose gel electrophoresis, and quantified using a Nanodrop ONE spectrophotometer (Thermo Scientific, Wilmington, DE, USA) prior to PCR. The 16S rRNA gene was amplified using the universal V3-V4 hypervariable region-specific fusion primers V3F (5′-ACGGHCCARACTCCTACGGAA-3′) and V4R (5′-CTACCMGGGTATCTAATCCKG-3′) using an ABI GeneAmp 9700 PCR thermocycler (Applied Biosystems, Foster City, CA, USA). Paired-end sequencing of the V3-V4 variable regions was performed on an Illumina MiSeq PE300 platform (Illumina, San Diego, CA, USA), according to standard protocols (Majorbio Bio-Pharm Technology Co., Ltd., Shanghai, China). Quality control of the raw 16S rRNA gene sequencing reads was performed using fastp version 0.20.0. Following quality screening, UPARSE version 7.1 was used to perform operational taxonomic unit (OTU) clustering of the sequences based on 97% similarity, and to eliminate chimeras. The taxonomy of each representative OTU sequence was determined using RDP Classifier version 2.2 against the 16S rRNA database, with a confidence threshold of 0.7. All experiments were conducted using three biological replicates.

### 4.8. Thrip Bioassay

This study used a modified thrip insecticide bioassay system (TIBS) approach, as described in Appendix A. Briefly, 10 mg each of coumestrol and 7,4′-dihydroxyflavone were dissolved in 0.1 mL of dimethyl sulfoxide (DMSO) to prepare the 100 mg/mL mother liquor. Both 7,4′-dihydroxyflavone and coumestrol were diluted to 4 mg/mL in distilled water. The control consisted of 4% DMSO. Twenty three-day-old adult female *M. usitatus* thrips were starved for 24 h and introduced into each 35 × 10 mm petri dish (contain 1% agarose). Then, 2 cm pods soaked in different concentrations for 30 s were put into petri dishes. The petri dish lid was covered with a 120-mesh nylon mesh then put onto the petri dish using parafilm seal, with 10 replicates per treatment. All tubes were maintained in an incubator (BIC-250, Shanghai Boxun Medical Biological Instrument Co., Ltd., Shanghai, China) at a temperature of 26 ± 0.5 °C, 60 ± 5% RH, and a photoperiod of 14:10 (L:D) h. After 48 h, the number of survivors was counted and the mortality was calculated. Thrips were considered dead if they were unable to move when disturbed with a soft brush.

### 4.9. Statistical Analysis

All data are presented as the mean ± standard error (SE). The data were analyzed by student *t*-test using SPSS (Version 13.0, SPSS Inc., Chicago, IL, USA) to determine statistical differences at the 5% level. Figures were produced using Origin (Version 2022b) software.

## 5. Conclusions

The formation of BHBT symptoms induced by thrip infestation was explored in this study. Herbivory by *M. usitatus* resulted in damage to the pod epidermis, changes to the cell wall structure, and increased accumulation of major cell wall components, markers of stress response, and phytohormones. It was worth noting that there was no significant difference in protein and free amino acid content between BHBT pods and healthy pods. *M. usitatus* was found to alter the bacterial community, but BHBT symptoms were formed without the involvement of pathogenic fungi or bacteria. Finally, *M. usitatus* induced the enrichment of flavonoid and isoflavonoid metabolites, including the two insecticidal compounds 7,4′-dihydroxyflavone and coumestrol. Our results shed light on the nature of BHBT symptoms in cowpea, and highlight two potential botanical pesticides for *M. usitatus* control.

## Figures and Tables

**Figure 1 plants-12-03865-f001:**
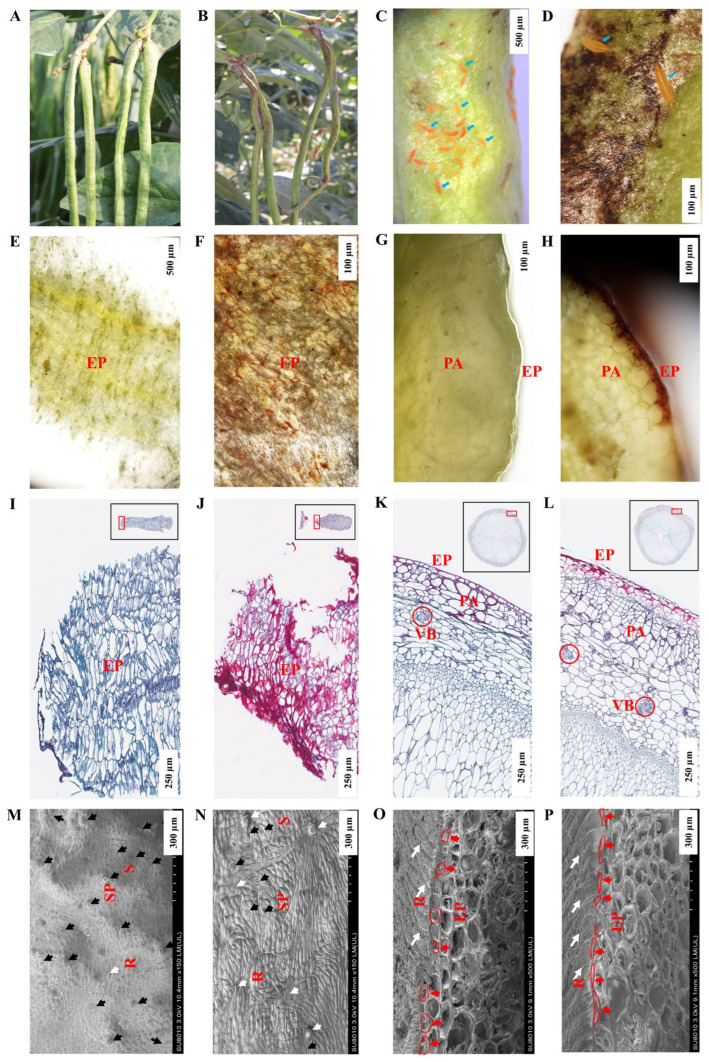
Morphology structure of healthy pods and BHBT pods. (**A**): Healthy pods, CK. (**B**): BHBT pods, T. (**C**): *M. usitatus* nymphs attacking pods, scale bar = 500 µm. (**D**): Cowpea pod of BHBT symptoms, scale bar = 100 µm. (**E**): Fresh section of CK cross section, scale bar = 500 µm. (**F**): Fresh section of BHBT, scale bar = 100 µm. (**G**): Fresh section of CK longitudinal section, scale bar = 100 µm. (**H**): Fresh section of BHBT, scale bar = 100 µm. (**I**): Paraffin section of CK cross section, scale bar = 250 µm. (**J**): Paraffin section of cross section of BHBT, scale bar = 250 µm. (**K**): paraffin section of longitudinal section of CK, scale bar = 250 µm. (**L**): paraffin section of longitudinal section of BHBT, scale bar = 250 µm. (**M**): SEM cross section of CK, scale bar = 300 µm. (**N**): SEM of cross section of BHBT, scale bar = 300 µm. (**O**): SEM of longitudinal section of CK, scale bar = 100 µm. (**P**): SEM of longitudinal section of BHBT, scale bar = 100 µm. EP: Epidermis. PA: Parenchyma cell. VB: Vascular bundle. R: Ridges. S: Stomatal pore. SP: Stomatal pore. Blue arrows indicate *M. usitatus* nymphs. Black arrows indicate stomatal locations. White arrows indicate damaged tissues. Red arrows indicate epidermal cells. Red boxes indicate the morphology of epidermal cells. CK: (**A**,**E**,**G**,**I**,**K**,**M**,**O**). T: (**B**–**D**,**F**,**H**,**J**,**L**,**N**,**P**).

**Figure 2 plants-12-03865-f002:**
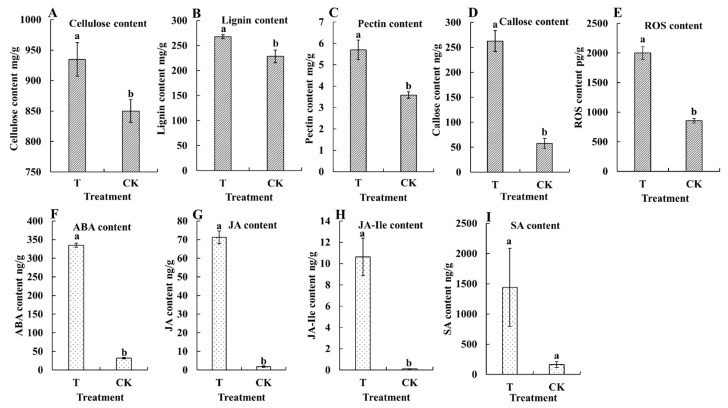
Contents of physiological and biochemical indexes in cowpea pods. (**A**): Cellulose content. (**B**): Lignin content. (**C**): Pectin content. (**D**): Callose content. (**E**): Reactive oxygen species (ROS) content. (**F**): Abscisic acid (ABA) content. (**G**): Jasmonic acid (JA) content. (**H**): Jasmonoyl–isoleucine (JA-Ile) content. (**I**): Salicylic acid (SA) content. Each value represents the mean ± SE of five replicates. Bars with different lower-case letters indicate significant differences based on a *t*-test at the *p* ≤ 0.05 level.

**Figure 3 plants-12-03865-f003:**
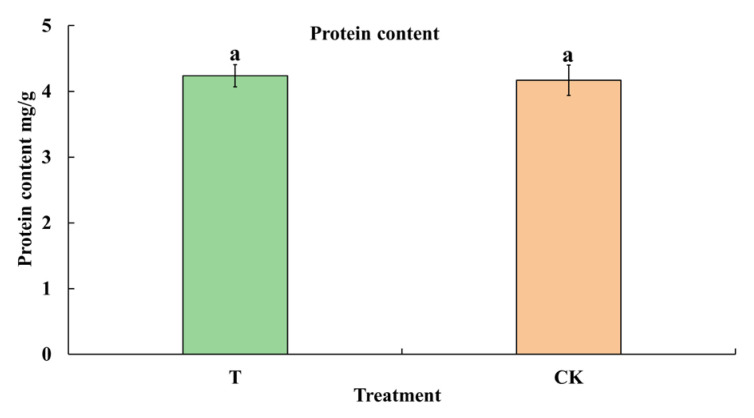
Protein content of healthy pods and BHBT pods. T: BHBT pods, CK: healthy pods. Each value represents the mean ± SE of five replicates. Bars with different lowercase letters indicate significant differences based on a *t*-test at the *p* ≤ 0.05 level.

**Figure 4 plants-12-03865-f004:**
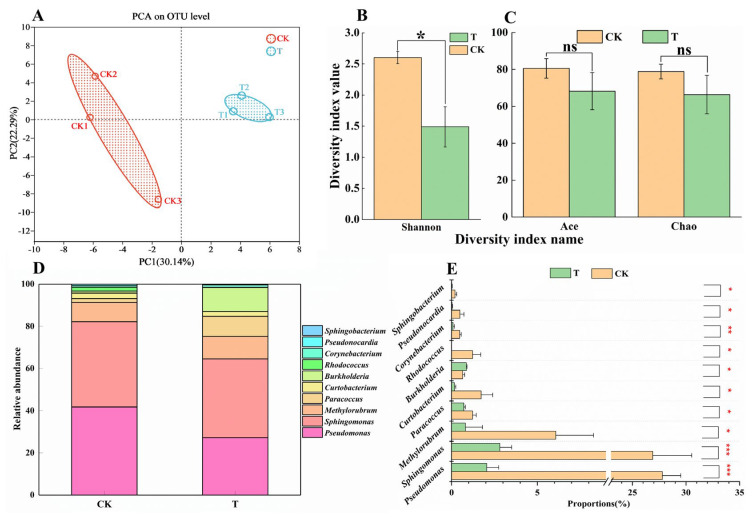
Analysis of bacterial community in cowpea pod. (**A**): Principal component analysis of OUT-level of epidermal bacterial communities in healthy pods (CK) and BHBT pods (T). (**B**,**C**): alpha diversity analysis at OUT-level of epidermal bacterial communities in healthy pods and BHBT pods. (**D**): The relative abundance analysis of epidermal bacterial communities in healthy pods and BHBT pods at the genus level. (**E**): The significant difference of epidermal bacterial communities at genus level between healthy pods and BHBT pods. The differences were considered significant based on a *t*-test at * *p* ≤ 0.05, ** *p* ≤ 0.01, *** *p* ≤ 0.001.

**Figure 5 plants-12-03865-f005:**
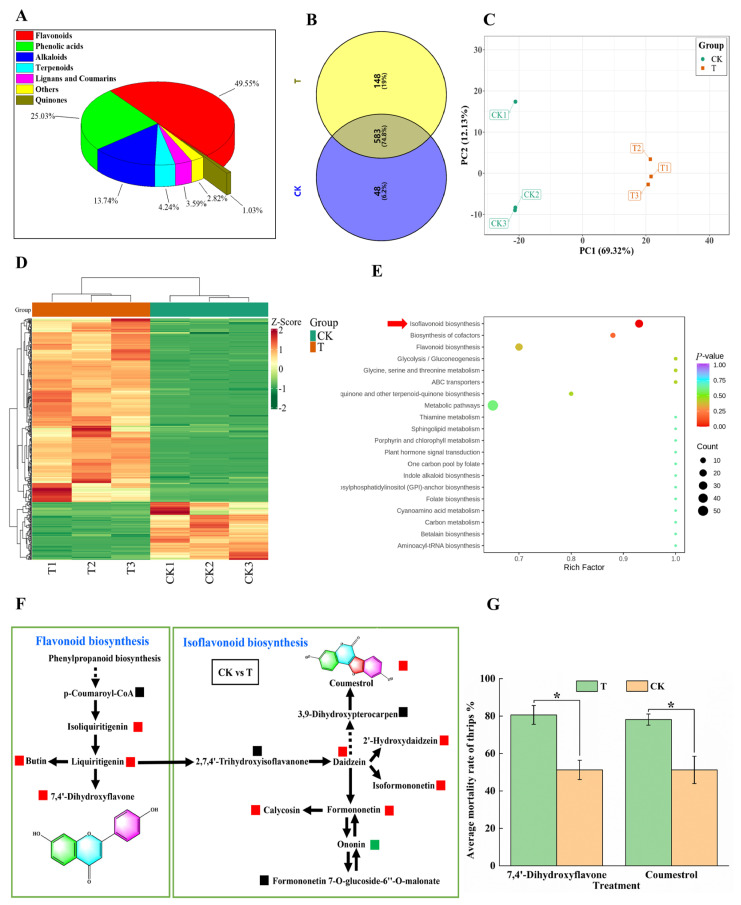
Specialized metabolites analysis of cowpea pods. (**A**): Composition of secondary metabolites in BHBT pod (T). (**B**): Venn diagram of secondary metabolites in healthy pods (CK) and BHBT pods. (**C**): Principal component analysis of secondary metabolites in healthy and BHBT pods. (**D**): Cluster analysis of secondary metabolites in healthy and BHBT pods. (**E**): Analysis of the enrichment pathway of secondary metabolites in healthy and BHBT pods. (**F**): KEGG biosynthesis pathway map of secondary metabolites in healthy and BHBT pods. Red indicates upregulation of specialized metabolites, green indicates downregulation of specialized metabolites, and black indicates that specialized metabolites are not detected. 7,4′-Dihydroxyflavone and coumestrol were highly upregulated in the flavonoid biosynthesis pathway and the isoflavonoid biosynthesis pathway. (**G**): bioassay of 7,4′-Dihydroxyflavone and coumestrol. T: The concentration is 4 mg/mL, CK: 4% DMSO. The differences were considered significant based on a *t*-test at * *p* ≤ 0.05.

**Table 1 plants-12-03865-t001:** The content of free amino acids in thrips herbivory pods and healthy pods.

Compound	T ^1^ (μg/g)	CK ^1^ (μg/g)	Foldchange	Log2FC	Up_Down	*p*-Value	Sig ^2^
Val	5.688474	6.265304	0.907933	−0.139340		0.543471	no
Gly	1.132279	1.640758	0.690095	−0.535130		0.000076	no
Ala	13.23861	6.051965	2.187490	1.129276	up	0.044021	yes
Ser	22.79589	14.80773	1.539459	0.622424		0.006758	no
Pro	8.507030	5.584443	1.523345	0.607242		0.101650	no
Thr	6.209276	8.046606	0.771664	−0.37396		0.014603	no
Ile	10.13251	9.572607	1.058490	0.082008		0.800122	no
Leu	11.30029	10.66793	1.059278	0.083081		0.807528	no
Asn	79.92255	79.55666	1.004599	0.006620		0.906895	no
Asp	53.37339	54.02035	0.988024	−0.01738		0.953402	no
Hcy	2.892285	2.956404	0.978312	−0.03163		0.237605	no
Gln	61.41007	84.6171	0.725741	−0.46247		0.001916	no
Lys	65.82454	91.84549	0.716688	−0.48058		0.002440	no
Glu	31.82564	27.71020	1.148517	0.199772		0.510139	no
Met	5.799168	8.766233	0.661535	−0.59611		0.017968	no
His	33.50499	38.20139	0.877062	−0.18925		0.282981	no
Phe	9.351112	9.565434	0.977594	−0.03269		0.822482	no
Arg	12.22800	11.52820	1.060704	0.085022		0.885041	no
Tyr	2.705990	3.206760	0.843839	−0.24496		0.258981	no
Trp	3.880709	3.601588	1.077500	0.107687		0.163580	no

^1^ T: Thrips herbivory pods, CK: Healthy pods. ^2^ The metabolites with log2FC ≥ 1 and *p*-value ≤ 0.05 were selected as the final differential metabolites. The differences were considered significant based on a *t*-test.

## Data Availability

The data presented in this study are available upon request from the corresponding author.

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
