# Peer review of "Megalurothrips usitatus Directly Causes the Black-Heads and Black-Tail Symptoms of Cowpea along with the Production of Insect-Resistance Flavonoids"

_plants, 2023, doi:10.3390/plants12223865_

Round 1

Reviewer 1 Report

Comments and Suggestions for Authors

This manuscript tackles disentangling the etiology of BHBT symptoms in cowpeas. The authors showed that the thrips attack is solely responsible for the occurrence of the symptoms. The presence of fungal spores was not detected in crosssections of BHBT-affected plants, and no pathogenic bacteria were identified in the cowpeas' microbiome. Elevation of expression of genes involved in plant defense was detected; among them, 7,4'-Dihydroxyflavone and coumestrol, for which the authors showed a direct effect on thrips. Also, this research pointed out that cowpeas with BHBT symptoms retained their protein and free AM content.

I really liked this research and the way it was crunched into a manuscript. It is well presented with clear and supported conclusions. I don't have any admonition. The tables and figures are informative and clear. Methods are informative and comprehensive. I particularly liked the conclusion regarding the nutritional value of BHBT-affected cowpeas. This kind of research should be done more often because it aligns with SDG goals (more quality, although not pretty food, for an ever-increasing number of humans).

Reviewer 2 Report

Comments and Suggestions for Authors

This article by He et al. investigates the mechanisms underlying the black-head black-tail (BHBT) symptoms induced by bean flower thrip (Megalurothrips usitatus) herbivory in cowpea. While BHBT causes major crop losses, the processes involved are poorly understood. This study provides comprehensive data on changes in pod morphology, biochemistry, metabolites, and microbiome after thrip damage. However, the manuscript would benefit from streamlining and better highlighting the most novel findings, as outlined below.

Major comments:

1) The introduction gives good background on cowpea as an important crop, the damage caused by bean flower thrip, and the characteristic black-head black-tail (BHBT) symptoms seen on pods. However, it would be strengthened by more clearly framing the gap in knowledge that this study aims to address. Specifically, the introduction would benefit from expanding on: What is currently known about the physiological and biochemical changes associated with thrip herbivory in cowpea or related species? What is understood about the mechanism of BHBT formation? Clearly setting up the gap in mechanistic understanding of BHBT formation would help establish the rationale and significance for the approaches taken in this study.

2) The results and discussion sections place heavy emphasis on analyzing changes in specialized metabolites like flavonoids and isoflavonoids after thrip herbivory. However, gaining a full picture of how thrips alter plant physiology also requires analyzing impacts on primary metabolism. Primary metabolites such as sugars, amino acids, and organic acids play crucial roles in plant growth, development, and defense responses. Levels of primary metabolites are often altered by insect feeding as the plant's metabolism is disrupted. Since the authors performed untargeted metabolomics, it would be informative to mention/discuss whether this data provided any insights into primary metabolic changes as well. What effects did thrip damage have on metabolites involved in photosynthesis, glycolysis, the TCA cycle, nitrogen assimilation, etc.

    3) The results demonstrate increased accumulation of the phytohormones ABA, JA, and JA-Ile in cowpea pods following thrip damage. The general roles of these hormones in insect defense have been introduced in the discussion. However, the significance of the specific changes observed in this study could be elaborated on further: Relate the magnitude of the increases seen (e.g. 2-fold for JA) to other studies examining insect herbivory. How do they compare? Explain the coordinated actions of ABA, JA, and JA-Ile in triggering anti-herbivory defenses. Expanding the discussion of the ABA/JA results would provide greater insight into the signaling mechanisms underlying cowpea's defense against bean flower thrips.

 Minor comments:

1)      Some of the figures contain scale bars that are difficult to visually interpret due to their small size.  I suggest increasing the thickness and font size of the scale bar numbers and lines in these figures to make them more readable. Enlarging the scale bars will allow readers to more easily assess dimensions and interpret the images appropriately.

2)      In Figure 4, ensure that all bacterial genus and species names are properly italicized for consistency with scientific nomenclature conventions. For example, Pseudomonas

3)      The legend for Figure 5F states that this shows the KEGG biosynthesis pathway map of secondary metabolites. However, the figure itself does not include much detail on the specific pathways depicted. Consider expanding the legend to provide more information. More descriptive figure legends help readers better understand the information presented without needing to refer extensively to the main text.

Round 2

Reviewer 2 Report

Comments and Suggestions for Authors

The authors have addressed all the comments.